

# Development of a preoperative index-based nomogram for the prediction of hypokalemia in patients with pituitary adenoma: a retrospective cohort study

Wenpeng Li[1,*], Lexiang Zeng[2,*], Deping Han[3], Shanyi Zhang[1], Bingxi Lei[1], Meiguang Zheng[1], Yuefei Deng[1] and Lili You[4]

[1] Neurosurgery, Sun Yat-sen Memorial Hospital, Sun Yat-sen University, Guangzhou, China
[2] Pediatric Surgery, Sun Yat-sen Memorial Hospital, Sun Yat-sen University, Guangzhou, China
[3] Neurosurgery, JieXi People's Hospital, JieXi, China
[4] Endocrinology, Sun Yat-sen Memorial Hospital, Sun Yat-sen University, Guangzhou, China
* These authors contributed equally to this work.

Corresponding authors
Yuefei Deng, 1249514956@qq.com
Lili You, youlli@mail.sysu.edu.cn

## ABSTRACT

**Objective:** To develop and validate a preoperative index-based nomogram for the prediction of hypokalemia in patients with pituitary adenoma (PA).

**Methods:** This retrospective cohort study included 205 patients with PAs between January 2013 and April 2020 in the Sun Yat-sen Memorial Hospital, Guangzhou, China. The patients were randomly classified into either a training set ($N = 143$ patients) and a validation set ($N = 62$ patients) at a ratio of 7:3. Variables, which were identified by using the LASSO regression model were included for the construction of a nomogram, and a logistic regression analysis was used to calculate odds ratios (ORs) and 95% confidence intervals (CIs) in the training set. The area under the curve (AUC) was used to evaluate the performance of the nomogram for predicting hypokalemia. Multivariate logistic regression analysis with a restricted cubic spline analysis was conducted to identify a potential nonlinear association between the preoperative index and hypokalemia.

**Results:** The incidence of hypokalemia was 38.05%. Seven preoperative indices were identified for the construction of the nomogram: age, type of PA, weight, activated partial thromboplastin time, urea, eosinophil percentage, and plateletocrit. The AUCs of the nomogram for predicting hypokalemia were 0.856 (95% CI [0.796–0.915]) and 0.652 (95% CI [0.514–0.790]) in the training and validation sets, respectively. Restricted cubic splines demonstrated that there was no nonlinear association between hypokalemia and the selected variables.

**Conclusion:** In this study, we constructed a preoperative indices-based nomogram that can assess the risk of hypokalemia after the surgical treatment of pituitary adenomas. This nomogram may also help to identify high risk patients who require close monitoring of serum potassium.

## INTRODUCTION

Pituitary adenomas (PAs) account for approximately 15% of all central nervous system tumors (*Ostrom et al., 2013*). The clinical manifestations of PAs include endocrine dysfunction (i.e. infertility, decreased libido and galactorrhea) and neurological deficiencies (i.e. headache and visual changes) (*Ezzat et al., 2004*; *Lake, Krook & Cruz, 2013*; *Melmed, 2015*). Surgery remains the treatment of choice for the majority of PAs (*Katznelson et al., 2014*; *Lampropoulos, Samonis & Nomikos, 2013*; *Nieman et al., 2015*). Serum potassium is one of the major intracellular components (*Ahmed et al., 2007*; *Hayes et al., 2012*; *Luo et al., 2016*; *Mattsson et al., 2016*; *Patel et al., 2017*; *Wang et al., 2013*). Hypokalemia is defined as levels of serum potassium <3.5 mmol/L (*Lodin & Palmér, 2015*). As many as 20% of hospitalized patients are observed to have hypokalemia, but only 4–5% of patients demonstrate specific clinical symptoms of hypokalemia. Undetected hypokalemia can lead to respiratory failure and is associated with morbidity and mortality (*Ashurst et al., 2016*; *Udensi & Tchounwou, 2017*; *Wojtaszek & Matuszkiewicz-Rowińska, 2013*). The most common cause of hypokalemia is drug induction, such as the induction of $\beta_2$-sympathomimetic drugs and diuretics, losses through the stool and kidney (due to infectious, tumors or malabsorption), and insufficient intake. Hypokalemia is a common feature in patients with adrenocorticotropic hormone (ACTH)-dependent Cushing's syndrome (CS) (80–85% of patients) (*Fan et al., 2020*; *Lacroix et al., 2015*). Furthermore, patients with thyroid-stimulating hormone (TSH)-secreting pituitary adenoma have reported incidences of hypokalemia (*Okuma et al., 2018*). We presumed that patients with pituitary adenomas may be prone to hypokalemia because of hormonal metabolism disorder and/or headache and/or nausea and/or vomiting.

Clinical parameters and preoperative routine biological indices including coagulation screening tests, routine biochemical examination, and routine hematology examinations at the time of admission of patients, are routine clinical practices and are standard practice that are performed before surgical procedures to evaluate a patient's overall general health and to assess preoperative risk. Preoperative clinical parameters and routine biological indices reflect the nutrition and metabolism of the body and are considered to be acceptable as variables for exploring the risk factors for hypokalemia.

Nomograms are widely used as assessment or prediction models for the prevalence and incidence of diseases (*Cen et al., 2020*; *Hu et al., 2019*; *Zhao et al., 2020*). A nomogram prognostic model for Small Cell Lung Cancer (SCLC) patients validated the model by using an independent patient cohort with an integrated area under the curve of 0.79 (*Wang et al., 2018*). Moreover, a nomogram model was established to identify the high-grade papillary bladder cancer and the area under the curve was 0.81 for internal validation and 0.78 for external validation (*Wakai et al., 2018*). However, to the best of our knowledge, no previous studies have investigated the significant preoperative factors that influence hypokalemia or have formulated prediction models of hypokalemia for PAs. We hypothesized that there are prediction indices in the preoperative period for hypokalemia and that we can take advantage of this knowledge to closely monitor serum potassium levels in certain patients to avoid hypokalemia, especially for asymptomatic patients with hypokalemia.

Therefore, this study aimed to develop and validate a preoperative nomogram model for the prediction of hypokalemia by combining clinical parameters and preoperative routine biological indices. The nomogram model can be conveniently used to facilitate the preoperative individualized prediction of hypokalemia in patients with PAs who are treated with surgery.

## METHODS

### Study location

This study was a retrospective study, that retrospectively collected information on medical records from the Department of Neurosurgery between January 2013 and April 2020 in the Sun Yat-sen Memorial Hospital.

### Inclusion and exclusion criteria

The following inclusion criteria were used: (1) patients with ages ≥ 18 years; (2) patients with Han descent from the Guangzhou area; and (3) patients who underwent surgery. The exclusion criteria were as follows: (1) patients with previous pituitary surgery; (2) patients who underwent selective adenectomy other than total tumorectomy; (3) subjects who received radiotherapy or chemotherapy before or after surgery; (4) patients who had preoperatively taken related drugs (within 3 months) that are associated with reducing serum potassium such as $\beta_2$-sympathomimetic drugs and diuretics; and, (5) subjects who were preoperatively diagnosed with enteritis via patients' self-reports or medical records within three months.

### Study population

A total of 221 consecutive PA patients were included in this study. All of the PA patients who participated in this study were treated with surgeries that were performed by experienced neurosurgeons. Patients who failed to provide postoperative serum potassium levels ($N = 16$ patients) were excluded from the analyses. Accordingly, a total of 205 eligible patients were included in the final data analyses. Patient characteristics and clinical details were obtained from medical records. All of the patients were of Han descent from the Guangzhou area.

### Ethics

The bioethics principles of the Declaration of Helsinki were strictly followed, and the study was approved by the Sun Yat-sen Memorial Hospital affiliated with Sun Yat-sen University (Reference Number: SYSEC-KY-KS-2020-118). The Institutional Review Board waived the need for consent because the study was based on the retrospective collection of information from medical records.

### Demographic and basic clinical data collection and measurement

Information on sociodemographic characteristics and basic clinical data regarding PAs were obtained from medical records. Current smoking and drinking habits were classified as "Yes" (indicating that patients were smoking or drinking regularly in the past 6 months)

or "No." Education levels were classified into elementary school and below, junior school, high school, and college degree or above. Preoperative information about the use of drugs (Yes/No), radiotherapy (Yes/No), or surgery (Yes/No) was obtained at admission by a neurosurgery physician. Body height and weight were measured by a nurse at admission to the nearest 0.1 cm and 0.1 kg, respectively, while patients were wearing light indoor clothing and without shoes. Body mass index (BMI) was calculated as the weight in kilograms divided by the height in meters squared (kg/m$^2$). Blood pressure measurements and heart rates were obtained by a nurse at admission, and patients were asked to rest while sitting for five minutes or longer.

Patients underwent diagnostic computed tomography (CT) scanning or magnetic resonance imaging (MRI) in the sellar region to determine preoperative tumor dimensions and type. PA classification was based on hormone immunohistochemical staining of the pituitary, including null cell adenomas (nonfunctioning), prolactinomas, ACTH adenomas, growth hormone adenomas, follicle-stimulating hormone adenomas, and thyroid-stimulating hormone adenomas. The histopathology examination was performed by following standard procedures and was immediately conducted for a confirmatory diagnosis of adenoma type.

### Evaluation of preoperative indices

Venous blood samples were collected for laboratory tests after overnight fasting or at least 10 h. All of the samples were obtained before 9:00 am. All of the collected samples were centrifuged within 30 min at 3,000 rpm for 10 min and were measured within 2 h. In this study, the preoperative biochemistry indices included preoperative routine coagulation, hematology, and biochemistry. The detailed variables are shown in Table S1. All of the measurement processes were precisely conducted according to the manufacturers' instructions. The variables in the preoperative routine coagulation measurement were detected on a fully automated coagulation analyzer Sysmex CS-5100 system TM (Siemens Healthcare Diagnostics, Erlangen, Germany). Preoperative routine hematology-related indices were measured on a Sysmex XN-9000 analyzer (Sysmex Corporation, Kobe, Japan). Finally, the variables in the preoperative routine biochemistry measurement were detected by using a Beckman Coulter AU 5800 (Beckman Coulter Inc., Brea, CA, USA).

### Definition of postoperative hypokalemia

Routine postoperative biochemistry evaluation was immediately conducted following surgery and was then repeated on the second postoperative day for all of the PA patients who underwent surgery. Normal serum potassium levels were defined as serum potassium levels ranging from 3.5 to 5.5 mmol/L. A serum potassium level <3.5 mmol/L was defined as hypokalemia (*Viera & Wouk, 2015*). In this study, we defined hypokalemia as a serum potassium level of <3.5 mmol/L on the day of surgery or on the second day after surgery. Serum potassium was measured by using an autoanalyzer (Beckman Coulter Inc., Brea, CA, USA).

## Statistical Analyses

With regard to missing data, the "missing at random" assumption was applied and multiple imputations with the multivariate normal imputation method were used (*Tsuchiya & Tsuchiya, 2019*). The numbers of patients with missing data for each of the variables are shown in Table S2. Variables missing at >20%, including uric acid nitrogen and creatinine (69.27%), apolipoprotein e (31.22%), CG (62.93%), LAP (21.95%), RBP (32.20%), iron saturation (70.73%), and SF (62.44%), were excluded before multiple imputation. Multiple imputations with chained equations ($m = 5$ imputed datasets) were conducted by using the mice R package. Following these procedures to obtain five complete datasets, a conventional statistical analysis was performed by using the first dataset. There was no significant difference between the original datasets and the complete dataset after multiple imputations (Table S3).

The continuous variables are presented as the mean ± standard deviation (SD) for normally distributed data, and a Student's *t*-test was used to compare the characteristics of the patients. In contrast, the variables with a skewed distribution are reported as medians (interquartile ranges). The differences between the groups were tested by using the Mann–Whitney *U* test. For the categorical variables, the data are presented as frequencies (percentages) and were compared by using the chi-square test.

The 205 patients who were enrolled in the study were divided into a training model set ($N = 143$ patients) and a validation model set ($N = 62$ patients) at a ratio of 7:3 by using a simple random sampling method. A least absolute shrinkage and selection operator (LASSO) logistic regression, which is suitable for the regression of high-dimensional data, was used to select the important preoperative indices for predicting postoperative hypokalemia in the training set, and the optimal values of the penalty parameter λ were determined via 10-fold cross-validation. A LASSO logistic regression analysis was conducted by using the glmnet R package. All logistic regression analysis was initiated began with the following clinical candidate predictors: age, type of PA, weight, activated partial thromboplastin time (APTT), serum potassium, urea, phosphocreatine kinase (CK), hematocrit (HCT), eosinophils (EOS), and plateletocrit (PCT). A backward stepwise selection was applied by using the likelihood ratio test with Akaike's information criterion as the stopping rule in multivariate logistic regression. Multivariate logistic regression models with a restricted cubic spline analysis were conducted to determine the potential nonlinear association between age, weight, APTT, serum potassium, urea, CK, HCT, EOS, and PCT (all of the continuous variables) and the risk of postoperative hypokalemia. To provide the clinician with a quantitative tool to predict the individual probability of hypokalemia, a preoperative index nomogram based on the LASSO analysis was established for further clinical application. The area under the curve (AUC) was calculated to evaluate the accuracy of the preoperative index predictive model for determining postoperative hypokalemia. The AUC value ranges from 1.0 to 0.5, with 1.0 indicating a perfect ability to correctly discriminate the outcome with the model and 0.5 indicating a random chance. The calibration plot and Hosmer–Lemeshow test were used to evaluate the calibration of the preoperative index prediction nomogram. We also performed

sensitivity analyses. We validated the nomogram model in a dataset without multiple imputations (raw data) and deleted the incomplete datasets in the variables that were selected by the LASSO regression (age, weight, type of pituitary adenomas, APTT, urea, EOS and PCT). We defined this dataset as a sensitivity analysis dataset. Statistical analyses were conducted by using RStudio software. All of the tests were two-tailed, and a two-sided $p$ value $< 0.05$ was considered to be statistically significant.

## RESULTS

### Basic characteristics of pituitary tumor patients

Seventy-eight out of the 205 pituitary tumor patients (38.05%) developed hypokalemia. The mean age was $47.28 \pm 14.26$ years. Of the 205 patients who were randomly divided into the training and validation sets at a ratio of 7:3, the prevalence of hypokalemia was 37.76% (54/134 patients) and 38.71% (24/62 patients), respectively ($p = 1.000$). There was no significant difference in the clinical and biochemical parameters between the training and validation sets (Table S4). The general characteristics of the patients with postoperative hypokalemia in the training and validation sets are shown in Table 1. Compared to patients without hypokalemia, those patients with hypokalemia were older and weighed less in the training set (all $p < 0.05$). The systolic blood pressures and heart rates were significantly lower in the hypokalemia patients than in the non-hypokalemia patients (all $p < 0.05$) in the validation set.

### Feature selection and preoperative index candidate building

The LASSO regression accurately identified important preoperative index candidates for predicting postoperative hypokalemia. After the LASSO regression analysis, 10 features with nonzero coefficients out of 103 preoperative indices were selected as being important indices for the prediction of postoperative hypokalemia (Figs. 1A and 1B).

### Association among preoperative indices and postoperative hypokalemia

The relationship between preoperative indices that were selected by the LASSO analysis and postoperative hypokalemia was confirmed with logistic regression analyses. As shown in Table 2, 10 selected preoperative indices, including age, type of PA, weight, APTT, serum potassium, urea, CK, HCT, EOS, and PCT, were closely associated with the prevalence of postoperative hypokalemia. In the multivariate logistic regression analyses, age was significantly associated with an increased risk of postoperative hypokalemia (odds ratio [OR], 1.06; 95% confidence interval [CI] [1.02–1.10]; $p = 0.002$) per unit increase. Weight (OR: 0.94; 95% CI [0.90–0.99]; $p = 0.010$), APTT (OR: 0.88; 95% CI [0.79–0.98]; $p = 0.019$), urea (OR: 0.52; 95% CI [0.35–0.74]; $p = 0.001$), and EOS (OR: 0.79; 95% CI [0.62–0.99]; $p = 0.050$) were significantly associated with a decreased risk of postoperative hypokalemia unit increase. The odds of postoperative increase were lower in functioning PAs than in nonfunctioning PAs (OR: 0.21; 95% CI [0.06–0.66]; $p = 0.011$). Compared with patients in the lowest quartile of PCT, the ORs in quartiles 2, 3, and 4 were 0.19 (95% CI [0.05–0.63]), 0.21 (95% CI [0.06–0.65]), and 0.10 (95% CI [0.03–0.37])

**Table 1 Characteristics of patients in the development and validation sets.**

| Variables | Total | Training dataset (N = 143) | | | | Validation dataset (N = 62) | | | |
|---|---|---|---|---|---|---|---|---|---|
| | | Non-hypokalemia (N = 89) | Hypokalemia (N = 54) | t/w/χ² | P | Non-hypokalemia (N = 38) | Hypokalemia (N = 24) | t/z/χ² | P |
| Male, n (%) | 104 (50.73) | 50 (56.17) | 24 (44.44) | 1.414 | 0.235 | 22 (57.89) | 8 (33.33) | 2.638 | 0.104 |
| Age, years | 47.30 ± 14.16 | 45.07 ± 13.60 | 51.20 ± 14.29 | −2.535 | 0.013 | 47.18 ± 15.60 | 47.00 ± 12.43 | 0.051 | 0.959 |
| Smoking, n (%) | 21 (10.24) | 12 (13.48) | 5 (9.26) | 0.240 | 0.624 | 3 (7.89) | 1 (4.17) | 0.003 | 1.000 |
| Drinking | 4 (1.95) | 1 (1.12) | 2 (3.70) | 0.195 | 0.557 | 1 (2.63) | 0 | <0.001 | 1.000 |
| Education | | | | | | | | | |
| Junior high school or below | 131 (63.90) | 52 (58.43) | 36 (66.67) | 0.647 | 0.421 | 30 (78.95) | 13 (54.17) | 3.164 | 0.075 |
| High school or above | 74 (36.10) | 37 (41.57) | 18 (33.33) | | | 8 (21.05) | 11 (45.83) | | |
| Marriage, n (%) | 177 (86.63) | 76 (85.39) | 49 (90.74) | 0.455 | 0.500 | 31 (81.58) | 21 (87.50) | 0.069 | 0.727 |
| Hypertension, n (%) | 136 (66.34) | 22 (24.72) | 21 (38.89) | 2.571 | 0.109 | 19 (50.00) | 7 (29.17) | 1.836 | 0.175 |
| Diabetes, n (%) | 39 (19.02) | 17 (19.10) | 13 (24.07) | 0.246 | 0.620 | 5 (13.16) | 4 (16.67) | 0.001 | 0.725 |
| Nonfunctioning pituitary adenomas | 163 (79.51) | 67 (75.28) | 48 (88.89) | 3.135 | 0.077 | 27 (71.05) | 21 (87.50) | 1.433 | 0.231 |
| Tumor diameter | 2.58 | (1.70–3.22) | 2.50 (1.70–3.22) | 2.51 | | (1.62–3.20) | 2505.5 | 0.671 | 2.60 |
| (1.90–3.40) | 2.36 | (1.68–3.22) | 0.771 | 0.444 | | | | | |
| Height, cm | 161.74 ± 11.60 | 162.48 ± 9.19 | 161.19 ± 11.36 | 0.707 | 0.481 | 160.92 ± 17.35 | 161.58 ± 9.19 | −0.196 | 0.845 |
| Weight, kg | 65.24 ± 11.17 | 66.77 ± 10.31 | 62.16 ± 9.72 | 2.683 | 0.008 | 65.10 ± 9.99 | 66.69 ± 16.98 | −0.416 | 0.680 |
| BMI, kg/m² | 25.58 ± 4.92 | 26.19 ± 4.45 | 24.84 ± 6.03 | 1.429 | 0.157 | 25.17 ± 4.19 | 25.68 ± 4.88 | −0.425 | 0.673 |
| SBP, mmHg | 124.93 ± 20.08 | 123.90 ± 19.89 | 125.98 ± 22.91 | −0.553 | 0.581 | 129.05 ± 17.79 | 119.88 ± 16.78 | 2.049 | 0.046 |
| DBP, mmHg | 80.70 ± 12.93 | 80.12 ± 13.81 | 79.50 ± 11.60 | 0.290 | 0.773 | 84.29 ± 11.96 | 79.88 ± 13.68 | 1.300 | 0.201 |
| Heart, times/minutes | 79.38 ± 13.24 | 79.66 ± 13.87 | 81.19 ± 14.44 | −0.620 | 0.537 | 79.53 ± 11.92 | 74.04 ± 8.40 | 2.124 | 0.038 |

Notes:
Data were mean ± SD or median (IQR) for skewed variables or numbers (proportions) for categorical variables.
P values were for the analysis of students' t test, Wilcox test or χ² analyses across the groups.
BMI, body mass index; SBP, systolic blood pressure; DBP, diastolic blood pressure.

for postoperative hypokalemia, respectively, and they showed a significant correlation with increased odds of postoperative hypokalemia.

## Restricted cubic splines of preoperative indices associated with the risk of postoperative hypokalemia

To more rigorously evaluate the association of preoperative indices that were selected by the LASSO regression with the risk of postoperative hypokalemia, logistic models with RCS were created. Nonlinear relationships were explored between the prevalence of postoperative hypokalemia and age (Fig. 2A), weight (Fig. 2B), APTT (Fig. 2C), serum potassium (Fig. 2D), urea (Fig. 2E), CK (Fig. 2F), HCT (Fig. 2G), EOS (Fig. 2H), and PCT (Fig. 2I). Multivariate logistic regression models with restricted cubic spline analyses
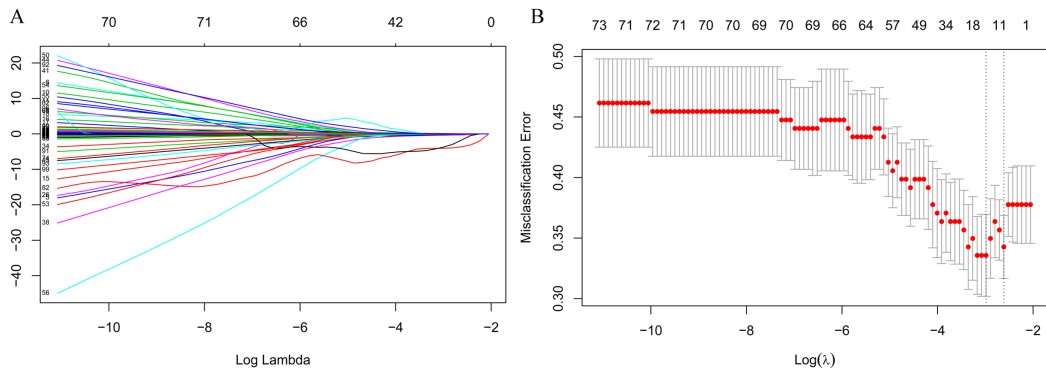

**Figure 1 Feature selection using the least absolute shrinkage and selection operator (LASSO) binary logistic regression model.** (A) Model LASSO. Color lines represent the preoperative indices associated with postoperative hypokalemia. The *x*-axis represents the alpha (cutoff), and the *y*-axis represents the shrink effect value. LASSO coefficient profiles of the 103 texture features. A coefficient profile plot was produced against the log (λ) sequence. A vertical line was drawn at the value selected using 10-fold cross-validation, where optimal λ resulted in 10 nonzero coefficients. (B) Tuning parameter (lambda) selection in the LASSO model used 10-fold cross-validation via minimum criteria for risk of postoperative hypokalemia. The area under the receiver operating characteristic (AUC) curve was plotted versus log (λ). Dotted vertical lines were drawn at the optimal values by using the minimum criteria. A λ value of 0.074, with log (λ), −2.603, was chosen (1 standard deviation) according to 10-fold cross-validation.

revealed that there was no nonlinear association of the selected preoperative indices, excluding CK (*p* for nonlinearity was 0.006) and EOS (*p* for nonlinearity was 0.039) with postoperative hypokalemia (all *p* < 0.05).

## Development and validation of an individualized prediction model for postoperative hypokalemia

LASSO regression analysis identified age, type of PA, weight, APTT, urea, EOS, and PCT as predictors for postoperative hypokalemia. The model for individualized prediction that incorporated the previously mentioned predictors was presented as the nomogram (Fig. 3). We constructed an ROC curve to evaluate the ability of the preoperative index prediction model to predict the prevalence of postoperative hypokalemia in the training set (Fig. 4A) and validation set (Fig. 4B). The AUCs of ROC in the training and validation sets were 0.856 (95% CI [0.796–0.915]) and 0.652 (95% CI [0.514–0.790]), respectively. The calibration curve of the preoperative nomogram for the prevalence of postoperative hypokalemia demonstrated good agreement (Figs. 5A and 5B), and the Hosmer–Lemeshow test yielded a nonsignificant statistic in the training (*p* = 0.189) and validation (*p* = 0.317) datasets.

### Sensitivity analyses

In the sensitivity analysis, 164 participants without missing information on variables in the nomogram model were included, including 63 participants who progressed to postoperative hypokalemia. We validated the nomogram model in the sensitivity analysis dataset and the results were consistent with the primary analyses (AUC: 0.795 (95% CI

**Table 2 Risk factors for postoperative hypokalemia in training dataset (N = 143).**

| Variables | Univariate model | | Multivariate model | |
|---|---|---|---|---|
| | OR (95% CI) | Ptrend | OR (95% CI) | Ptrend |
| Age, years | 1.03 [1.0–11.06] | 0.013 | 1.06 [1.02–1.10] | 0.001 |
| Functioning pituitary adenoma | 0.38 [0.13–0.96] | 0.052 | 0.21 [0.06–0.66] | 0.011 |
| Weight, kg | 0.96 [0.92–0.99] | 0.011 | 0.94 [0.90–0.99] | 0.010 |
| APTT, S | 0.91 [0.84–0.98] | 0.012 | 0.88 [0.79–0.98] | 0.019 |
| Serum potassium, mmol/L | 0.21 [0.06–0.63] | 0.007 | | |
| Urea, mmol/L | 0.79 [0.61–1.00] | 0.056 | 0.52 [0.35–0.74] | 0.001 |
| CK, U/L | 0.988 [0.981–0.996] | 0.003 | | |
| HCT | | | | |
| ≤0.370 | 1.00 | 0.019 | | |
| (0.370, 0.396) | 0.33 [0.13–0.85] | | | |
| (0.396, 0.429) | 0.45 [0.18–1.13] | | | |
| >0.429 | 0.24 [0.08–1.68] | | | |
| EOS, % | 0.82 [0.68–0.98] | 0.041 | 0.79 [0.62–0.99] | 0.050 |
| PCT, % | | | | |
| ≤0.22 | 1.00 | 0.064 | 1.00 | 0.001 |
| (0.22, 0.26) | 0.29 [0.11–0.77] | | 0.19 [0.05–0.63] | |
| (0.26, 0.31) | 0.44 [0.18–1.07] | | 0.21 [0.06–0.65] | |
| >0.31 | 0.37 [0.13–0.98] | | 0.10 [0.03–0.37] | |

Notes:
Data are ORs (95% CI). Participants without postoperative hypokalemia are defined as 0 and with postoperative hypokalemia as 1.
Backward step-wise selection was applied by using the likelihood ratio test with Akaike's information criterion as the stopping rule in multivariate logistic regression.
APTT, activated partial thromboplastin time, S; CK, phosphocreatine kinase, U/L; HCT, hematocrit; EOS, eosinophils percentage, %; PCT, plateletocrit, %.

[0.727–0.864])) and the Hosmer–Lemeshow test yielded a nonsignificant statistic ($p = 0.926$). The results are shown in Table S5.

# DISCUSSION

By selecting routinely used clinical parameters and preoperative biological indices of hospitalized PA patients, the results of this study suggest that patient age and type of PA, including nonfunctioning and ACTH types, were associated with an increased risk of postoperative hypokalemia. Lower weight, as well as APTT, urea, EOS and PCT levels were also significantly associated with an increased risk of postoperative hypokalemia. Furthermore, we developed and validated a preoperative index-based nomogram prediction model for postoperative hypokalemia in PA patients. The nomogram incorporated seven items of preoperative indices, including, age, type of PA, weight, APTT, urea, EOS, and PCT.

The prevalence of postoperative including in patients with PAs was 37.07% (76/205 patients), which was higher than the reported prevalence of hypokalemia (20%) in general hospitalized patients. Moreover, Marti et al. showed that the prevalence of hypokalemia in patients presenting to the emergency department was 11%, which was considerably lower

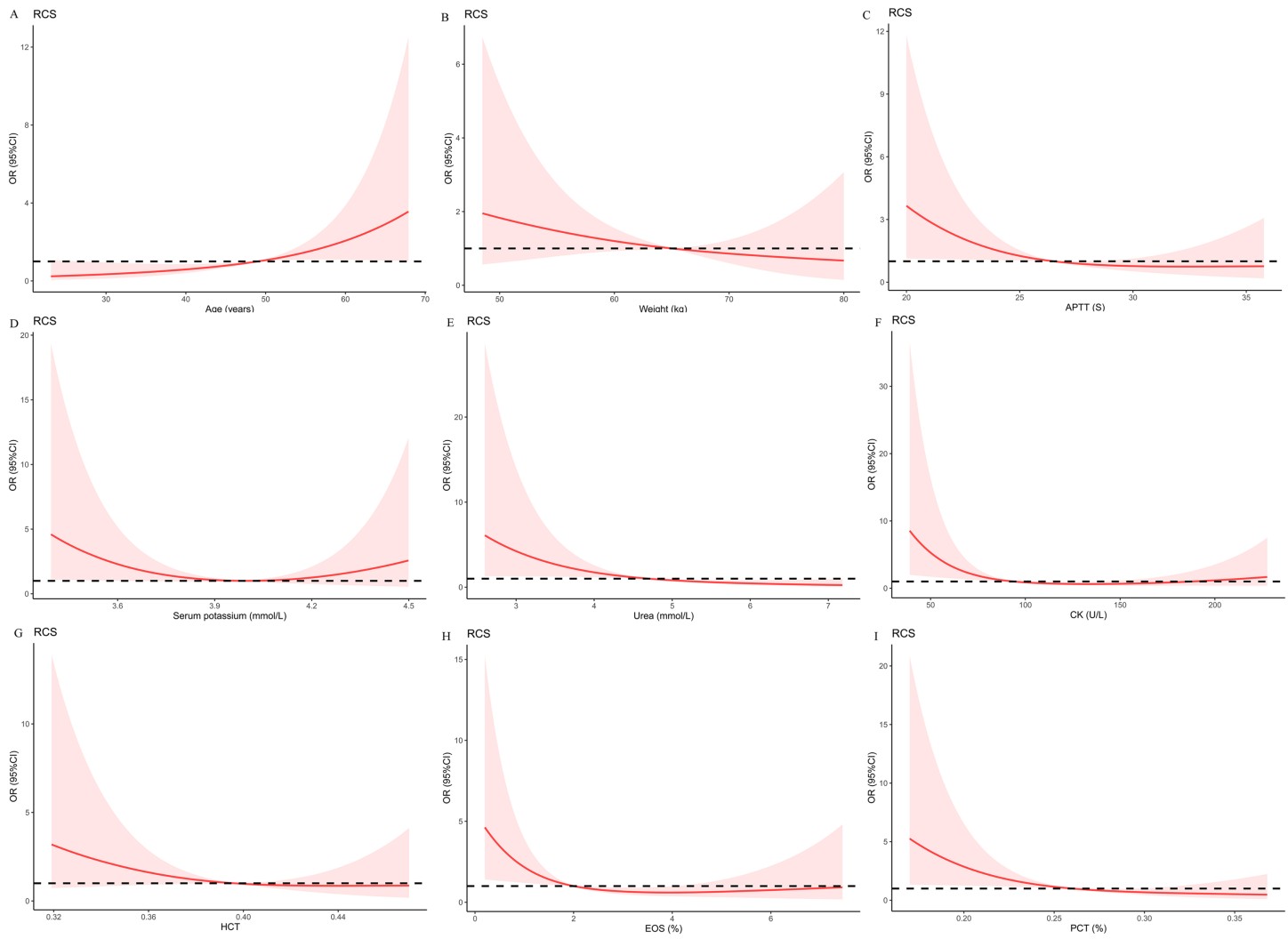

**Figure 2 Multivariable logistic regression models restricted cubic spline analysis the selected variables by LASSO regression analyses on a continuous scale and risk for prevalence postoperative hypokalemia.** Odds ratios are represented by a solid line, and the 95% confidence intervals are represented by the pink area. (A) to (I) were restricted cubic spline modeling of the association of age, weight, APTT, serum potassium, urea, CK, HCT, EOS, and PCT with risk of postoperative hypokalemia adjusted by types of pituitary adenomas and the other eight indices.

than our findings (38.05%). *Hamad et al. (2019)* reported that the prevalence of hypokalemia in peritoneal dialysis patients was 34.0%, which was similar to our study. We suggest that the prevalence of hypokalemia in patients with PAs who undergo surgery is more common than that in general hospitalized patients; therefore, these cases should receive more attention from doctors.

Similar to other studies (*Nilsson et al., 2017*; *Sarafidis et al., 2012*), we found that age was associated with an increased risk of postoperative hypokalemia. Reported that each 10 year increase in age was associated with a 40% higher risk of hypokalemia. Furthermore, a cohort study based on 364,955 participants concluded that older age was associated with a higher risk of hypokalemia (*Nilsson et al., 2017*). Additionally, *Kleinfeld et al. (1993)* reported a significant correlation of the prevalence of hypokalemia with increased age

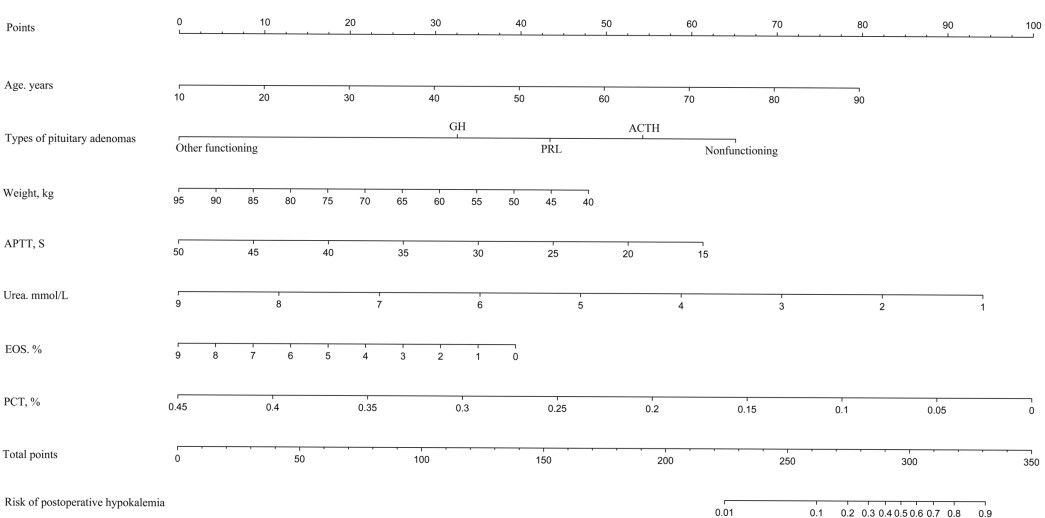

**Figure 3 Developed and validated preoperative indices nomogram.** The preoperative indices nomogram was developed with age, type of pituitary adenoma, weight, APTT, serum potassium, urea, CK, HCT, EOS, and PCT.

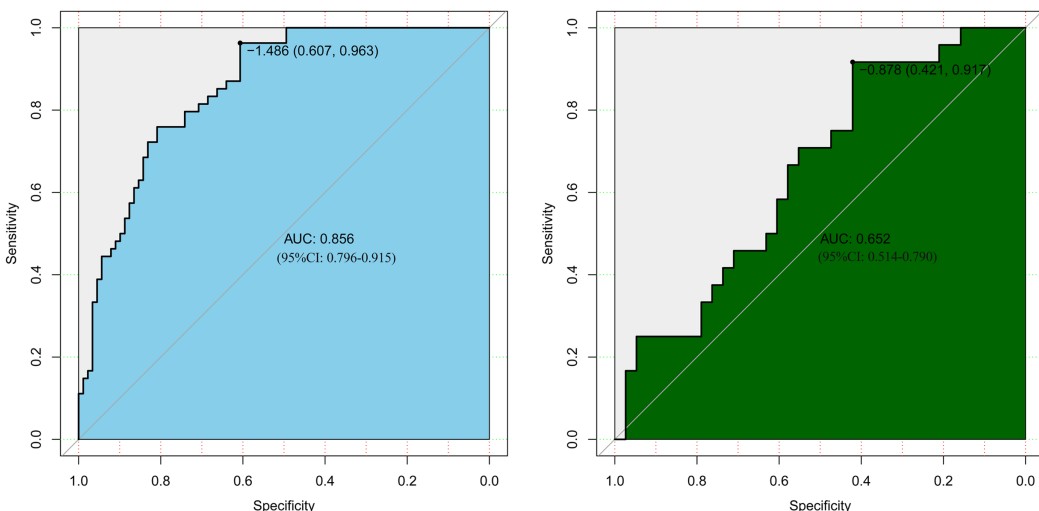

**Figure 4 Receiver operating characteristic curves (ROC) for the preoperative index's prediction model predicting postoperative hypokalemia in training and validation sets.** AUC, area under the curve.

($p < 0.001$). We believe that the associations of hypokalemia and age disparity are clear. Moreover, recent evidence has demonstrated that this association was possibly due to age-related differences in body mass composition, which results in physiologically low total exchangeable body potassium in older patients (*Kleinfeld et al., 1993*). Many researchers have also concluded that the type of PA is associated with a difference in the prevalence of hypokalemia (*Fan et al., 2020*; *You et al., 2017*). *You et al. (2017)* reported that, compared with other specific types of PA, patients with ACTH-PAs had a significantly higher prevalence of postoperative hypokalemia. ACTH-secreting PAs are related to a clinical disorder known as Cushing's disease (CD), and hypokalemia is a common

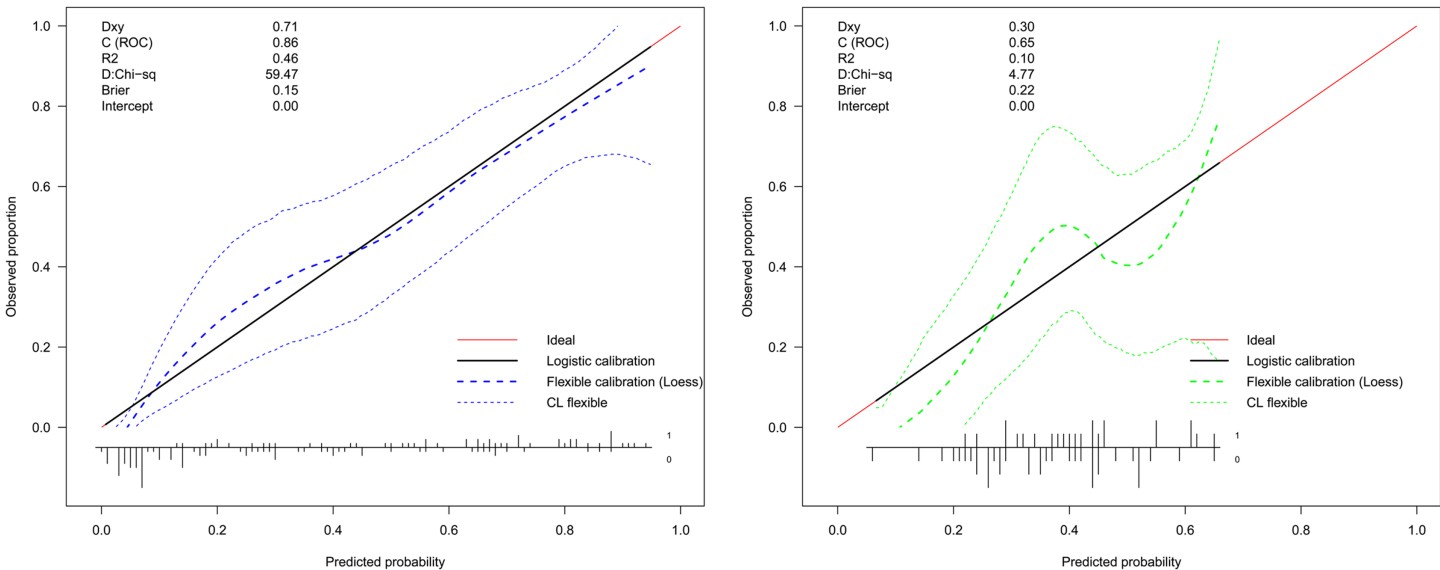

**Figure 5 Calibration curve of the preoperative indices' nomogram.** The *y*-axis represents the actual postoperative hypokalemia rate. The *x*-axis represents the predicted postoperative hypokalemia risk. The diagonal dotted represents a perfect prediction by an ideal model. The dotted line represents the performance of the nomogram, of which a closer fit to the diagonal dotted line represents a better prediction.

feature in patients with CD. In these patients, a high cortisoll level may be the principal cause of hypokalemia (*Fan et al., 2020*). It is noteworthy that our study first reported that nonfunctioning PAs were associated with a high risk of postoperative hypokalemia. There are two possible explanations for these findings. (1) The diameter of the tumor for patients with nonfunctioning PAs receiving surgery was larger than that for functioning PAs (nonfunctioning PAs vs. functioning PAs: 2.70 [2.10–3.41] vs. 1.20 [0.80–2.45], respectively; *p* < 0.001), and the mass effect of the pituitary gland on surrounding structures was obvious. The mass effect of the pituitary gland on surrounding structures was more significantly reduced or completely eliminated in a shorter period of time after surgery in nonfunctioning PAs compared to functioning PAs, which can cause obvious hormone level variations in the body, can further trigger adaptation functions in the body, and may lead to postoperative ion metabolism disorder. (2) There were only a small number of patients with functioning PAs in this study, which may have had a disproportionate influence on parameter estimates for the association of PA subtypes and postoperative hypokalemia.

Variables including weight and low levels of urea reflect a nutritional status of inadequate protein intake. Our study suggests that preoperative malnutrition was closely associated with an increased prevalence of postoperative hypokalemia. Reported that hypokalemia is associated with poor nutritional status in ambulatory peritoneal dialysis patients. Patients with malnutrition increased risk of hypokalemia because of insufficient intake, excessive loss and electrolyte distribution (*Hortencio et al., 2016*). Furthermore, low levels of APTT, EOS, and PCT reflecting the status of coagulopathy or anemia were significantly associated with postoperative hypokalemia in this study. Importantly, anemia

is one of the biomarkers for malnutrition that is associated with hypokalemia, and anemia and hypokalemia often appear at the same time for acute or traumatic diseases (*Shigemi et al., 2011*; *Stewart, Traylor & Bratzke, 2015*). In a study including 112 patients with traumatic disease, half of the patients developed increased coagulative activity and progressive anemia at an early stage, after which 60% of the patients presented with hypokalemia (*Khudaĭberenov, 1979*). Preoperative indices related malnutrition and anemia may change the levels of pH and Na+, as well as change the osmolality and K +-ATPase activity which can shift the internal distribution of K+ ions. Preoperative monitoring of malnutrition- and anemia-related indices may be useful for predicting postoperative hypokalemia.

Certain limitations to this study should be considered. First, the limited number of patients who were selected from a single center should be noted. Furthermore, the relatively small number of patients does not allow us to investigate the relative risks for postoperative hypokalemia for every subtype of PA. The association of the PA subtype with postoperative hypokalemia should be further explored with larger, multicenter, cohort studies. Additionally, due to the retrospective design of the study, the reasons for postoperative hypokalemia were not clear and may have biased our results. However, we constructed a prediction nomogram model according to preoperative indices, which included clinical parameters and routine preoperative biological indices for predicting the risk of postoperative hypokalemia. Third, detailed information, such as the ambulation statuses of patients and the oral intake of potassium, could not be extracted from the medical records; again, this may have biased our results. However, we defined postoperative hypokalemia on the basis of levels of serum potassium on the day of surgery and the first postoperative day. During this time, most patients were bedridden and fasting. Thus, we do not believe that the ambulation status of patients or the oral intake of potassium has an important impact on the occurrence of postoperative hypokalemia. Moreover, our model requires further external validation before being applied for general use. However, our model is a first step in creating highly accurate tools for predicting postoperative hypokalemia.

## CONCLUSION

The prevalence of postoperative hypokalemia is high in patients with PAs who underwent surgery. Only age, type of PA, weight, APTT, urea, EOS, and PCT were significantly associated with postoperative hypokalemia in patients with PAs who underwent surgery. Furthermore, our study presents a preoperative indices nomogram that incorporates both clinical parameters and preoperative biological indices, including the items that were previously mentioned. We believe that this can be conveniently used to facilitate an individualized prediction of postoperative hypokalemia among PA patients. Our study is the first to emphasize the importance of paying clinical attention to postoperative hypokalemia and it is the first study to establish a nomogram prediction model according to preoperative indices for postoperative hypokalemia. Our study provided evidence to encourage the monitoring of postoperative serum potassium and supplementation in patients with PAs who are undergoing surgery.

## ACKNOWLEDGEMENTS

We are indebted to the participants in the present study for their outstanding support and to our colleagues for their valuable assistance.

### Funding

The authors received no funding for this work.

### Competing Interests

The authors declare that they have no competing interests.

### Author Contributions

- Wenpeng Li conceived and designed the experiments, performed the experiments, analyzed the data, prepared figures and/or tables, authored or reviewed drafts of the paper, and approved the final draft.
- Lexiang Zeng conceived and designed the experiments, performed the experiments, analyzed the data, prepared figures and/or tables, authored or reviewed drafts of the paper, and approved the final draft.
- Deping Han performed the experiments, prepared figures and/or tables, authored or reviewed drafts of the paper, colocted rawdatas, and approved the final draft.
- Shanyi Zhang performed the experiments, prepared figures and/or tables, authored or reviewed drafts of the paper, materials, and approved the final draft.
- Bingxi Lei performed the experiments, prepared figures and/or tables, authored or reviewed drafts of the paper, and approved the final draft.
- Meiguang Zheng performed the experiments, prepared figures and/or tables, authored or reviewed drafts of the paper, materials, and approved the final draft.
- Yuefei Deng conceived and designed the experiments, performed the experiments, analyzed the data, prepared figures and/or tables, authored or reviewed drafts of the paper, and approved the final draft.
- Lili You conceived and designed the experiments, performed the experiments, analyzed the data, prepared figures and/or tables, authored or reviewed drafts of the paper, and approved the final draft.

### Ethics

The following information was supplied relating to ethical approvals (i.e., approving body and any reference numbers):

The bioethics principles of the Declaration of Helsinki were strictly followed, and the study was approved by the Sun Yat-sen Memorial Hospital affiliated to Sun Yat-sen University (Reference Number: SYSEC-KY-KS-2020-118).

### Data Availability

The raw measurements are available in the Supplemental File.
## Supplemental Information

Supplemental information for this article can be found online at http://dx.doi.org/10.7717/peerj.11650#supplemental-information.

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
