# Peer review of "Development of a preoperative index-based nomogram for the prediction of hypokalemia in patients with pituitary adenoma: a retrospective cohort study"

_PeerJ, doi:10.7717/peerj.11650_

## Round 0.1 · original submission · Major Revisions

The paper has some issues, please address them accordingly.

·

Basic reporting

This manuscript is well writing and holds an important topic of discussion. I have no serious concerns against this draft, however; I would like to suggest few changes before considering this draft for publication. Authors to consider to modify the draft according to suggested changes in methodology and discussion section.

Experimental design

Can author explain that whey they opted for 7:3 distribution for training and validation cohort. Would it be not better to have equal distribution?
The definition of the hypokalemia widely varies across studies, authors must provide sound justification of using such cut off values for the purpose of current study.
Please provide appropriate references for definition section in the methodology.
Authors have provided much details in the statistical section, even study participants features and distribution is discussed in this section. Authors may consider to make following headings in the methodology section 1. ethics, 2. study location. 3. study population, 4. inclusion and exclusion criteria, 5. data collection and management, 6. statistics

Validity of the findings

The findings of the current study is interesting. Author tried to relate these findings with the other studies, but still rigor in comparison is currently lacking. Authors should describe the association and possible mechanism of predictive variable with the hypo-K, following its corroborating and contravening comparison with other studies.
Please also state the normal values of AUC in the statistics section that what is the meaning of high and low AUC.

Additional comments

Manuscript requires few corrections in grammar in syntax, particularly in introduction and discussion section.
Authors should consider to reduce the word count in the introduction section. this section is quite verbatic. This section should primarily discuss the linkage between PAs and hypo-k, and its impact on prognosis and outcomes. Moreover, rational of the study must be clearly enumerated in this section, e.g. why this study is need of time in the presence of previous literature.

Reviewer 2 ·

Basic reporting

This study aimed to evaluate the impact of clinical parameters and preoperative biological indices on postoperative hypokalemia and to build and validate predictive models for postoperative hypokalemia with the significant parameters. The structure of this article had an acceptable format of standard sections, the logicality of the article was also relatively fluent.

Experimental design

1) In the introduction, it should be introduced whether there are similar models in this field, and if so, how accurate the prediction of such models is.
2)It was recommended to use non-random grouping in the grouping of training set and validation set, so that the difference between groups could help test the external authenticity of the model. Authors could refer to the TRIPOD statement (PMID: 25561516).
3)The authors used restricted cubic spline modeling to analyze the independent variables and found that some independent variables were not monotonically increasing or decreasing in relation to the outcome. However, in the final Nomogram model, those independent variables were still used as continuous variable, such as serum potassium. Thus the significance of the application of restrictive cubic subsection was lost, and the model would be not very accurate. The author should treat the above variables according to the results of the restricted cubic spline.
4)Authors should refer to the Tripod and Probast statement. Transparent reporting of a multivariable prediction model for individual prognosis or diagnosis (TRIPOD): the TRIPOD statement. The TRIPOD Group (PMID:25561516), and PROBAST: A Tool to Assess Risk of Bias and Applicability of Prediction Model Studies: Explanation and Elaboration (PMID: 30596876).

Validity of the findings

no comment

Additional comments

It is suggested that the author modify the construction method of the model and the analysis method of independent variables according to the above suggestions to re-estimate the prediction effect of the model.

Reviewer 3 ·

Basic reporting

Detailed comments as below.

Experimental design

Detailed comments as below.

Validity of the findings

Detailed comments as below.

Additional comments

1. Line 62-63, it is inappropriate to describe both ROC and AUC are used to assess the performance, because AUC is an indicator and ROC is a method which can generate AUC.

2. Given the small sample size of validation set and the lack of external validation set, the conclusion should be made with caution. It is overstated to say “can facilitate the prediction of hypokalemia”.

3. Please follow the TRIPOD statement to write this paper. For this revision, please attach the corresponding TRIPOD list.

4. Introduction. It remains unclear why hypokalemia is a concern of postoperative patients with pituitary adenoma, and a further question is why preoperative could be used to predict postoperative hypokalemia. These are important for this research topic. The authors should also briefly review risk factors and reasons for postoperative hypokalemia in patients with pituitary adenoma.

5. Line 130, please specify the design of this study.

6. Line 130-141, please provide detailed inclusion and exclusion criteria.

7. A fatal methodological limitation is the small sample sizes, limiting the generalizability of the predictive model developed. Please also provide the 95%CI of AUC, due to this reason, I speculate the lower limit of AUC is lower than recommended.

8. For addressing missing values, please provide rationales for the use of multivariate normal imputation method. It seems no supporting evidence for random missing. The authors may consider to repeat the analysis with the complete dataset to do a sensitivity analysis.

---

## Round 0.2 · Major Revisions

I do not think the paper was carefully edited although the authors provided the language editing certificate. For example, line 114 "espectively" and line 107 "builded". Please re-edit the paper all throughout.

Reviewer 2 ·

Basic reporting

Clear and unambiguous, professional English were used throughout in this manuscript. Literature references, sufficient field background/context provided.

Experimental design

Research question well defined, relevant & meaningful. It is stated how research fills an identified knowledge gap.

Validity of the findings

All underlying data have been provided; they are robust, statistically sound, & controlled.

Additional comments

This study aimed to develop and validate a preoperative indices-based nomogram for the prediction of postoperative hypokalemia in patients with pituitary adenoma (PA). 205 patients were retrospectively collected in this study, and a nomogram prediction model was trained and validated. The AUCs of the nomogram model for predicting postoperative hypokalemia were 0.871 and 0.753 in the training and validation sets, respectively. This suggests that the authors have built a relatively good prediction model.
Overall, the study was well designed and the usage of statistical analysis was comprehensive. But there are small problems:
1) Variables that are not statistically significant in Table 2, such as serum potassium and CK, are not recommended to be included in Nomogram.
2) It is recommended that the report of the manuscript refer to TRIPOD "Transparent reporting of a multivariable prediction model for individual prognosis or diagnosis (TRIPOD): the TRIPOD statement". And the PROBAST (prediction model risk of bias assessment tool) was recommended to evaluate the risk of bias for this manuscript.

---

## Round 0.3 · accepted · Accept

My concerns are also addressed.

Reviewer 3 ·

Basic reporting

The authors have addressed my concerns. I have no fruther comments.

Experimental design

The authors have addressed my concerns. I have no fruther comments.

Validity of the findings

The authors have addressed my concerns. I have no fruther comments.

Additional comments

The authors have addressed my concerns. I have no fruther comments.